# Maternal Inherited Thrombophilia in Monochorionic Twin Pregnancy with Twin-Twin Transfusion Syndrome

**DOI:** 10.3390/jcm11237054

**Published:** 2022-11-29

**Authors:** Stefano R. Giannubilo, Alessia Fiorelli, Daniela Marzioni, Giovanni Tossetta, Giulia Capogrosso, Andrea Ciavattini

**Affiliations:** 1Department of Clinical Sciences, Polytechnic University of Marche, Salesi Hospital, Via Corridoni 11, 60123 Ancona, Italy; 2Department of Experimental and Clinical Medicine, Polytechnic University of Marche, Via Tronto, 10/a, 60126 Ancona, Italy

**Keywords:** monochorionic, placenta, thrombophilia, TTTS, twins

## Abstract

Background: To study the frequency of inherited thrombophilia in monochorionic twin pregnancies with twin-twin transfusion syndrome (TTTS). Methods: At the Department of Obstetrics of the Polytechnic University of Marche (Ancona, Italy) a population of monochorionic diamniotic pregnant women was selected retrospectively. After termination of the pregnancy, genotyping for Factor I, Factor V Leiden, Factor II and Methylenetetrahydrofolate Reductase (MTHFR), as well as activities of the plasma proteins C and S, was performed. Results: Regarding the 32 patients with TTTS, from a cohort of 104 monochorionic pregnancies recruited, at least one thrombophilic defect was more frequent (OR: 3.24), and the allele polymorphism frequency was higher for Factor I (OR: 4.4) and for Factor V Leiden (OR: 11.66). Conclusions: Maternal inherited thrombophilia, possibly also inherited from monochorial fetuses, may result in impaired development of the placental vascular architecture. This inheritance hypothesis may explain why only a fraction of monochorionic diamniotic twins develop TTTS.

## 1. Introduction

Monochorionic twin pregnancy is associated with an increased risk of fetal mortality and morbidity resulting from placental vascular complications [1]. The most common complications are twin-twin transfusion syndrome (TTTS), twin anemia polycythemia sequence, selective intrauterine growth restriction (IUGR) and single and double intrauterine fetal death (IUFD), compared to dichorionic twins [2,3,4]. The twin-twin transfusion syndrome occurs in 10–20% of monozygous gestations [5] and is categorized by the Quintero Classification, which aids in the diagnosis and formulation of management plans [6]. From the first description in 1687, the origin and pathophysiology of TTTS are still unknown. It is associated with 70% of the fetal mortality rate without treatment [7,8] and 10–30% of the neurological handicaps of the twin survivor after the co-twin’s intrauterine death [9]. Many different treatment strategies have been proposed and practiced, but fetoscopic laser photocoagulation, first described by De Lia et al. in 1990 [10], is currently the primary treatment modality pursued [11]. The monochorionic placenta may be considered a developmental malformation. The placental architecture of inter-twin vascular communication has been the subject of several studies, such as the vascular diameter, vascular resistance and flow patterns [12]. Twin-to-twin transfusion may develop from several vascular anastomoses between the circulations of each twin, with a subsequent preferential blood flow from one twin (donor) to the other one (recipient). However, placental histopathology suggests that vascular anastomosis is present in 100% of monochorionic twin pregnancies, but twin-twin transfusion affects <10% of them, probably because the majority do not have major arterial-venous anastomoses. Monochorionic twins present also a higher incidence of fetal vessel thrombosis compared to dichorionic [13]. Major forms of thrombophilia with a hereditary basis are the consequence of a G1691A mutation in Factor V Leiden, which may be homozygous or heterozygous. Thrombophilia is a condition of an increased tendency to form blood clots. People who have thrombophilia are at greater risk of having thromboembolic complications, such as deep venous thrombosis, pulmonary embolisms and cardiovascular complications, such as a stroke or myocardial infarction. Thrombophilias have been implicated in a variety of adverse obstetric events, including pregnancy loss (especially fetal death), preeclampsia, small-for-gestational-age infants and placental abruption [14]. Anticoagulant therapy has the potential to improve the obstetric outcome in women with heritable thrombophilia [15,16]. The most prominent placental lesions in women with thrombophilia and adverse pregnancy outcomes are fetal stem vessel thrombosis, infarcts, hypoplasia, spiral artery thrombosis and perivillous fibrin deposition [17]. The overall prevalence of thrombophilia is one in ten in the general population [18]. Nevertheless, there are no data in the literature regarding the frequency of hereditary thrombophilia in a population of monochorionic twin pregnancies. The aim of the present study was to retrospectively compare the frequency of hereditary thrombophilic factors in monochorionic diamniotic twin pregnancies relative to the incidence of TTTS.

## 2. Materials and Methods

Patients with monochorionic diamniotic pregnancies, who were followed or delivered at the Department of Obstetrics and Gynecology of the Polytechnic University of Marche, Salesi Hospital (Ancona, Italy) between January 2014 and March 2020, were enrolled in a case–control study. The study group was, in turn, subdivided into those with or without (controls) TTTS. Medical records were reviewed to obtain baseline information and perinatal outcomes. Chorionicity was defined on sonographic criteria [19,20] and was first determined by the presence of the T sign in the first trimester of pregnancy. The exclusion criteria included all other types of twins, pregnancies with fetal congenital structural abnormalities detected antenatally or postnatally and those who received anticoagulant therapy. According to the Eurofetus criteria, the TTTS definition criteria are polyhydramnios (deepest pocket ≥ 8 cm before 20 weeks of gestation or ≥10 cm after 20 weeks of gestation) in the recipient and oligohydramnios (deepest pocket ≤ 2 cm) in the donor [21]. The TTTS staging was performed according to the Quintero criteria [6]. All the patients recruited were asked to participate in the study and subjected to a blood sampling at least six months after the end of pregnancy. The activities of the plasma proteins C and S were measured using a functional clotting assay (Instrumentation Laboratory, Milano, Italy), and the measurements were performed in women out of pregnancy and without hormone intake. The cut-off for deficiency was a protein C activity of 76% and a protein S activity of 56%. A non-enzymatic technique was used to isolate the DNA from the whole blood. Genotyping was performed by polymerase chain reaction (PCR) using specific primers as previously described for the 455 G > A polymorphism in the Factor I gene, the G20210 G > A polymorphism in the Factor II gene, the 1691 G > A polymorphism in the Factor V gene [22,23] and the C677T polymorphism for the methylenetetrahydrofolate reductase (MTHFR) gene [24]. The PCR product variants were digested by specific restriction enzymes and analyzed on 1.5% agarose. The statistical analysis was performed using the Statistical Package for Social Sciences (SPSS) v. 19.0 (IBM Inc., Armonk, NY, USA). The data are shown as the means ± standard deviation (SD) or number (percentage), and the statistical differences between groups were tested with Student’s *t*-test. The chi-square test was utilized to assess the differences in frequencies. The results of the genotyping analysis are presented as the odds ratio (OR) or as the mean difference, with a 95% confidence interval (CI).

## 3. Results

### Subject Characteristics

A total of 104 patients with monochorionic twin pregnancies were recruited; the characteristics of the groups with and without TTTS are given in Table 1. The TTTS patients were comparable for the maternal age, pre-pregnancy body mass index and nulliparity frequency. As expected, the gestational age at delivery and birthweight were lower in the TTTS group (*p* < 0.001). The fetal demise of both twins was more frequent in the TTTS group (25% vs. 5.5%; *p* < 0.05). Deficiencies of free protein S were detected in only two cases in the TTTS group; no cases of protein C deficiency were detected. At least one thrombophilic defect was observed in 11 (34.3%) cases and 10 (13.8%) controls (OR: 3.24; *p* < 0.05) (Table 2). A total of 22 (68.7%) patients were found positive for Factor I (Beta-Fibrinogen) in the TTTS group, and 20 (62.5%) were heterozygous in the TTTS group versus 22 (30.5%) in the controls (OR: 3.78; *p* < 0.05). At least one mutation for Factor V Leiden was found in eight (25%) TTTS patients and in two2 (2.7%) controls (OR: 11.66; *p* < 0.05). A prothrombin mutation was detected as heterozygous in five (15.6%) cases in the TTTS group and one case (2.7%) in the controls. The difference in the frequency of the MTHFR C6777 homozygous mutation was not significantly different among the two groups. The coexistence of at least two defects was found not significantly different between the study group and the controls (31.2% vs. 25%), while the presence of at least one defect was more frequent in the study group than in the control group. No significant correlation was found between the thrombophilic mutations and the Quintero Stage in the TTTS patients group.

## 4. Discussion

Monochorionic twin pregnancies have a higher risk of complications compared to dichorionic pregnancies because most placentas have vascular anastomoses that connect the two fetal circulations and, through these anastomoses, the twins continuously exchange blood. The pathogenesis of TTTS remains incompletely understood; while the vast majority of diamniotic-monochorionic twin gestations have inter-twin anastomoses, only a fraction of them develops TTTS. Twin pregnancy itself has an alteration of the coagulation state; enhanced coagulability in twin pregnancies, expressed as decreased levels of fibrinogen and an increased level in the D-dimer, was previously reported [25]. To our knowledge, this is the first report of the frequency of hereditary thrombophilic factors in monochorionic twin pregnancies, and we found that at least one thrombophilic factor is more frequent in a population of patients who experienced TTTS, and the most frequent polymorphisms concern the Factor I and Factor V Leiden. Although thrombophilia is extensively studied for its implications in pregnancy complications, the role of different factors has not been completely elucidated, nor has it been given enough relevance to fetal thrombophilia in the etiology of those complications. The fetuses only have a 50% chance of inheriting maternal thrombophilia because it is transmitted in an autosomal dominant fashion; this may explain why not all pregnancies appear to be affected. In the case of a monochorionic pregnancy, both fetuses would carry the defect inherited, at least from the mother (even if the father is not a carrier). The placental tissue carries the same genotype as the fetus; this implies that, when the fetoplacental unit is affected by the genotype, the two sides of thrombophilia (maternal and fetal) may coincide [26], resulting in impaired development of the placental vascular architecture. Khong et al. reported the case of one dizygotic twin who inherited thrombophilic genes from both the father and mother, resulting in placental fetal thrombotic vasculopathy and intrauterine growth restriction, whereas its co-twin inherited only one such gene from its mother and was unaffected [27]. On the fetus-neonatal side, moreover, TTTS in monozygous twins is associated with an independent risk of perinatal arterial stroke (OR: 31.2, 95% CI: 2.9 to 340.0) [28]. At least one genetic risk factor for thrombophilia (protein C deficiency, Factor V Leiden or Factor II G20210A) was identified in half of the infants with a stroke at birth [29]. In this study, we found that some inherited thrombophilic factors were more frequent in the women who had a monochorionic pregnancy with TTTS than without, according to the hypothesis that a hereditary thrombophilia placental vascular pathology could be the pathophysiological link between inherited thrombophilia, pregnancy complications and neonatal thrombosis.

## 5. Conclusions

Our findings, even if in a small sample population, may encourage future larger studies with an aim to examine the possible link between inherited thrombophilia and placental vascular growth in monochorionic twin pregnancies. Understanding the mechanisms regulating coagulation within the placenta will be very important in order to establish preventive therapeutic options, for example, the use of low-molecular-weight heparin or low-dose aspirin, such as in other pathologies of pregnancy [30].

## Figures and Tables

**Table 1 jcm-11-07054-t001:** Characteristics of study groups.

	WITHOUT-TTTS (N = 72)	TTTS (N = 32)	*p*
Maternal age (years)	32.2 ± 3.7	31.7 ± 2.3	N.S.
Pre-pregnancy Body Mass Index	22.8 ± 6.2	23.1 ± 4.7	N.S.
Nulliparous	31 (86.1)	13 (81.2)	N.S.
Assisted reproduction	4 (5.5)	0	N.A.
Gestational age at delivery	33.1 ± 1.1	29.3 ± 3.1	<0.001
Birthweight larger twin (g)	2106 ± 431	1183 ± 348	<0.001
Birthweight smaller twin (g)	1923 ± 505	936 ± 377	<0.001
Cesarean section	68 (94.4)	32 (100)	N.S.
Fetal demise one twin	2 (2.7)	2 (6.2)	N.S.
Fetal demise both twins	4 (5.5)	8 (25)	<0.05
Quintero Stage I		4 (12.5)	
Quintero Stage II		12 (37.5)	
Quintero Stage III		4 (12.5)	
Quintero Stage IV		4 (12.5)	
Quintero Stage V		8 (25)	

Data are expressed as Mean ± Standard Deviation or N and (%). N.S.: not significant.

**Table 2 jcm-11-07054-t002:** Polymorphism frequency analysis.

Thrombophilic Factor	Genotype	WITHOUT-TTTS (N = 72)	TTTS (N = 32)	OR (95% CI)	*p*
Factor I 455G > A	AA	2 (2.7%)	2 (6.2%)	2.33 (95% CI: 0.3139 to 17.3470)	0.407
	GA	22 (30.5%)	20 (62.5%)	3.78 (95% CI: 1.5808 to 9.0765)	0.002 *
	GA + AA	24 (33.3%)	22 (68.7%)	4.4 (95% CI: 1.8000 to 10.7554)	0.001 *
Factor V 1691G > A	AA	0	2 (6.2%)	11.8 (95% CI: 0.5541 to 254.9433)	0.113
	GA	2 (2.7%)	6 (18.7%)	8.07 (95% CI: 1.5320 to 42.5827)	0.013 *
	GA + AA	2 (2.7%)	8 (25%)	11.6667 (95% CI: 2.3149 to 58.7990)	0.002 *
Factor II 20210G > A	AA	0	0	N.A.	N.A.
	GA	2 (2.7%)	5 (15.6%)	5.00 (95% CI: 0.8662 to 28.8617)	0.072
	GA + AA	2 (2.7%)	5 (15.6%)	5.00 (95% CI: 0.8662 to 28.8617)	0.072
Protein S deficiency		0	2 (6.2%)	7.06 (95% CI: 0.2725 to 183.1753)	0.2391
Protein C deficiency		0	0	N.A.	N.A.
MTHFR C677T	TT	24 (33.3%)	16 (50%)	2.0 (95% CI: 0.8559 to 4.6732)	0.109
At least one defect		10 (13.8%)	11 (34.3%)	3.24 (95% CI: 11.2078 to 8.7321)	0.019 *
Multiple defects		18 (25%)	10 (31.2%)	1.36 (95% CI: 0.5443 to 3.4161)	0.508

Numbers, percentages and odds ratios (OR) with their 95% confidence intervals (CI) are given; (*) *p* < 0.05; N.A.: not applicable.

## Data Availability

M.O. had access to the complete dataset used in the study and takes responsibility for the integrity of the data and accuracy of the data analyses. The dataset is available upon justified request.

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
