# Peer review of "Maternal Inherited Thrombophilia in Monochorionic Twin Pregnancy with Twin-Twin Transfusion Syndrome"

_jcm, 2022, doi:10.3390/jcm11237054_

Round 1
Reviewer 1 Report
The study aims to describe the prevalence of thrombophils in monochorionic pregnancy complicated by TTTS.
The manuscript needs a major revision since erroneous data has been detected in table 2, in the TTTS column... the data in this column are the same as the data in the same column of table 1. This makes difficult the interpretation of the results.
Besides, the structure of the discussion should be improved in:
- results of this study.
-Comparison of the results with the literature data-
- Clinical significance
- Limitations of the study
- Conclusions
Finally, review the typographic mistakes like :
- Pag1, line 28: space in “andmorbidity”
- Pag1, line 30: elective => selective.
Author Response
Point 1: The manuscript needs a major revision since erroneous data has been detected in table 2, in the TTTS column... the data in this column are the same as the data in the same column of table 1. This makes difficult the interpretation of the results.
Response 1: The TTTS column of table 2 has been changed by inserting the correct data
Point 2: Besides, the structure of the discussion should be improved in: - results of this study; Comparison of the results with the literature data; Clinical significance; Limitations of the study; Conclusions
Response 2: The discussion section has been rewritten according to the suggestions of the Reviewer
Point 3: Finally, review the typographic mistakes like : - Pag1, line 28: space in “andmorbidity” - Pag1, line 30: elective => selective.
Response 3: The typographyc mistakes have been correct as suggested

Reviewer 2 Report
The manuscript is interesting, made on a unique biological material.
I think it is necessary to structure the materials and methods. The methods of genetic variants detection should be described more clearly.
It is necessary to modify the table 2 and indicate the number of patients with different genotypes in both groups, the genotype frequency (there are three genotypes for each biallelic gene variant) in accordance with the genetic nomenclature (http://varnomen.hgvs.org/). The results also need to be adjusted.
The discussion should be shortened. In my mind discussion will be more interesting if you add more results of genetic studies of pregnancy twin pathologies.
Author Response
Point 1: I think it is necessary to structure the materials and methods. The methods of genetic variants detection should be described more clearly.
Response 1: The methods section has been modified as suggested
Point 2: It is necessary to modify the table 2 and indicate the number of patients with different genotypes in both groups, the genotype frequency (there are three genotypes for each biallelic gene variant) in accordance with the genetic nomenclature (http://varnomen.hgvs.org/). The results also need to be adjusted.
Response 2: As suggested by the Reviewer, the table has been modified by inserting the correct nomenclature for the allelic variants of the genes studied
Point 3: The discussion should be shortened. In my mind discussion will be more interesting if you add more results of genetic studies of pregnancy twin pathologies.
Response 3: The discussion section has been rewritten according to the suggestions of the Reviewer.

Round 2
Reviewer 1 Report
Thanks for the changes suggested in the revision
Author Response
no request for changes was received by the Reviewer 1

Reviewer 2 Report
It is necessary to replace "alleles" with "genotype" in Table 2
Author Response
As suggested, in the Table 2, the word "alleles" has been replaced with "genotype"
